# Public preferences for delayed or immediate antibiotic prescriptions in UK primary care: A choice experiment

Liz Morrell[1]*, James Buchanan[1,2,3], Laurence S. J. Roope[1,2,3], Koen B. Pouwels[1,2], Christopher C. Butler[2,4], Benedict Hayhoe[5], Sarah Tonkin-Crine[2,4], Monsey McLeod[6,7,8], Julie V. Robotham[9], Alison Holmes[6], A. Sarah Walker[2,3,10], Sarah Wordsworth[1,2,3], STEPUP team[¶]

1 Health Economics Research Centre, Nuffield Department of Population Health, University of Oxford, Oxford, United Kingdom, 2 NIHR Health Protection Research Unit in Healthcare Associated Infections and Antimicrobial Resistance, University of Oxford, Oxford, United Kingdom, 3 NIHR Biomedical Research Centre Oxford, John Radcliffe Hospital, University of Oxford, Oxford, United Kingdom, 4 Nuffield Department of Primary Care Sciences, University of Oxford, Oxford, United Kingdom, 5 Department of Primary Care and Public Health, School of Public Health, Imperial College London, London, United Kingdom, 6 NIHR Health Protection Research Unit in Healthcare-Associated Infections and Antimicrobial Resistance, Imperial College London, London, United Kingdom, 7 Centre for Medication Safety and Service Quality, Pharmacy Department, Imperial College Healthcare NHS Trust, London, United Kingdom, 8 NIHR Imperial Patient Safety Translational Research Centre, Imperial College London, London, United Kingdom, 9 Modelling and Economics Unit, National Infection Service, Public Health England, London, United Kingdom, 10 Nuffield Department of Medicine, John Radcliffe Hospital, University of Oxford, Oxford, United Kingdom

¶ Membership of STEP-UP team is provided in the Acknowledgements.
* liz.morrell@ndph.ox.ac.uk

## Abstract

### Background

Delayed (or "backup") antibiotic prescription, where the patient is given a prescription but advised to delay initiating antibiotics, has been shown to be effective in reducing antibiotic use in primary care. However, this strategy is not widely used in the United Kingdom. This study aimed to identify factors influencing preferences among the UK public for delayed prescription, and understand their relative importance, to help increase appropriate use of this prescribing option.

### Methods and findings

We conducted an online choice experiment in 2 UK general population samples: adults and parents of children under 18 years. Respondents were presented with 12 scenarios in which they, or their child, might need antibiotics for a respiratory tract infection (RTI) and asked to choose either an immediate or a delayed prescription. Scenarios were described by 7 attributes. Data were collected between November 2018 and February 2019. Respondent preferences were modelled using mixed-effects logistic regression.

The survey was completed by 802 adults and 801 parents (75% of those who opened the survey). The samples reflected the UK population in age, sex, ethnicity, and country of residence. The most important determinant of respondent choice was symptom severity,

**Data Availability Statement:** Data cannot be shared publicly because the terms of our ethics clearance included that the data would be held on secure networks at our University. Data are

available from the Information Governance Committee, Nuffield Department of Population Health, University of Oxford (contact via ig. support@ndph.ox.ac.uk) for researchers who meet the criteria for access to confidential data.

**Funding:** The support of the Economic and Social Research Council (UK) is gratefully acknowledged (https://esrc.ukri.org) [ES/P008232/1]. This study was also funded by the National Institute for Health Research (NIHR, https://www.nihr.ac.uk) Health Protection Research Unit in Healthcare Associated Infections and Antimicrobial Resistance at the University of Oxford in partnership with Public Health England (PHE, https://www.gov.uk/ government/organisations/public-health-england) [HPRU-2012-10041]. ASW and CCB are NIHR Senior Investigators. ASW, JB, LSJR and SW are supported by the NIHR Oxford Biomedical Research Centre (https://oxfordbrc.nihr.ac.uk/). MM is supported by the NIHR Imperial Patient Safety Translational Research Centre (https://www. imperial.ac.uk/patient-safety-translational-research-centre). The funders played no role in the design of the study, data collection and analysis, decision to publish, or preparation of the manuscript.

**Competing interests:** The authors have declared that no competing interests exist.

**Abbreviations:** CI, confidence interval; DCE, discrete choice experiment; GP, general practitioner; NHS, National Health Service; OR, odds ratio; RTI, respiratory tract infection; SD, standard deviation.

especially for cough-related symptoms. In the adult sample, the probability of choosing delayed prescription was 0.53 (95% confidence interval (CI) 0.50 to 0.56, $p < 0.001$) for a chesty cough and runny nose compared to 0.30 (0.28 to 0.33, $p < 0.001$) for a chesty cough with fever, 0.47 (0.44 to 0.50, $p < 0.001$) for sore throat with swollen glands, and 0.37 (0.34 to 0.39, $p < 0.001$) for sore throat, swollen glands, and fever. Respondents were less likely to choose delayed prescription with increasing duration of illness (odds ratio (OR) 0.94 (0.92 to 0.96, $p < 0.001$)). Probabilities of choosing delayed prescription were similar for parents considering treatment for a child (44% of choices versus 42% for adults, $p = 0.04$). However, parents differed from the adult sample in showing a more marked reduction in choice of the delayed prescription with increasing duration of illness (OR 0.83 (0.80 to 0.87) versus 0.94 (0.92 to 0.96) for adults, $p$ for heterogeneity $p < 0.001$) and a smaller effect of disruption of usual activities (OR 0.96 (0.95 to 0.97) versus 0.93 (0.92 to 0.94) for adults, $p$ for heterogeneity $p < 0.001$). Females were more likely to choose a delayed prescription than males for minor symptoms, particularly minor cough (probability 0.62 (0.58 to 0.66, $p < 0.001$) for females and 0.45 (0.41 to 0.48, $p < 0.001$) for males). Older people, those with a good understanding of antibiotics, and those who had not used antibiotics recently showed similar patterns of preferences. Study limitations include its hypothetical nature, which may not reflect real-life behaviour; the absence of a "no prescription" option; and the possibility that study respondents may not represent the views of population groups who are typically underrepresented in online surveys.

## Conclusions

This study found that delayed prescription appears to be an acceptable approach to reducing antibiotic consumption. Certain groups appear to be more amenable to delayed prescription, suggesting particular opportunities for increased use of this strategy. Prescribing choices for sore throat may need additional explanation to ensure patient acceptance, and parents in particular may benefit from reassurance about the usual duration of these illnesses.

## Author summary

### Why was this study done?

- Antibiotic resistance is a growing threat to global public health, and reduction of unnecessary antibiotic consumption is essential.

- An effective strategy to reduce antibiotic prescribing in primary care is delayed prescribing, where the patient is given a prescription but told to "wait and see" and take the antibiotics if their condition gets worse or does not improve; however, despite supporting evidence from randomised trials, this approach is not widely used in UK primary care.

- As doctor's prescribing decisions can be influenced by factors such as patient concerns, our study aimed to understand which factors affect patients' acceptance of delayed prescription and the relative importance of these factors.

## What did the researchers do and find?

- We designed an online choice experiment that had 12 possible situations a patient might encounter if they went to see their doctor about a respiratory tract infection (RTI). For each, we asked whether they would prefer an immediate or a delayed antibiotic prescription.

- A total of 802 adult members of the public and a further 801 adults who were parents of at least 1 child under 18 completed the survey.

- The most important features affecting prescription preference were the symptoms and how long they (or their child) had been ill. Respondents were most likely to choose the delayed prescription for minor symptoms like a common cold (probability of 53%) or minor sore throat (47%) and less likely for a serious chest infection (30%); only 37% chose delayed prescription for a sore throat, swollen glands, and fever, although this is likely to be a viral illness.

- We identified groups of respondents—for example, females and people who are knowledgeable about antibiotics—who were the most amenable to delayed prescription.

## What do these findings mean?

- The general public have some understanding that immediate antibiotics are not needed for colds, but better education on their (lack of) a role in viral sore throats may be helpful.

- Primary care doctors wishing to use more delayed prescription, but concerned about patient acceptance, could increase use among the more amenable groups.

- Patients will need reassurance that delayed prescription is appropriate for the more serious symptoms such as fever and on the typical duration of these illnesses.

## Introduction

Antibiotic resistance is a global threat to public health [1], and there is a need for improved stewardship of existing antibiotics [1,2]. Delayed (or "backup") antibiotic prescription in primary care is an approach to reducing antibiotic consumption. The patient is given a prescription, but advised to only initiate antibiotics if the condition worsens or does not improve within a specified time frame. This approach has been shown in clinical trials to be effective in reducing consumption of antibiotics without increasing complication rates relative to "standard" immediate prescriptions, in respiratory tract infections (RTIs) [3–7], urinary tract infections [8], and conjunctivitis [9]. However, despite UK primary care guidelines now recommending the use of delayed prescription [10,11], uptake has been modest in the UK and Europe [6,7,12]. It is therefore important to understand the factors that might be limiting the uptake of this effective prescribing option to help increase its use.

In the UK's National Health Service (NHS), prescribing decisions for antibiotics in primary care is generally made by a primary care physician (a general practitioner, GP). Besides clinical

features, the prescribing decision may be influenced by other factors [13], including the patient's concerns and expectations [14]; hence, it is important to understand the preferences of patients for antibiotic prescribing.

We address this question in the context of an RTI, one of the most common reasons for visits to primary care [10,15] and a major reason for (unnecessary) primary care antibiotic prescribing [3,16–18]. We focus on sore throat and cough-related symptoms, which account for a large proportion of unnecessary prescribing [19,20] and for which delayed prescription has been shown to be effective with little negative impact on clinical outcomes [4,6,7].

Our study aimed to understand which factors affect public preferences for a delayed rather than an immediate prescription and quantify the trade-offs that the public are prepared to make between them. As it is possible that preferences for antibiotics are different when parents are seeking treatment for their child, rather than for themselves, we conducted our study in 2 UK populations: adults and parents of at least 1 child under 18 years.

## Methods

The study used a stated preference survey approach. We presented respondents with 12 hypothetical situations they might encounter when consulting a primary care physician about an RTI and asked them to choose the type of prescription they would prefer in each one. The situations were described in terms of 7 attributes of the illness and the clinical consultation. By describing the situations as multiattribute profiles, we reflect the complexity of real healthcare decisions. We tested the hypothesis that each of these attributes affects respondent choices and quantified their relative importance.

Study design, data collection, and analysis followed good practice guidance for choice-based studies [21]. Ethical approval was granted by the University of Oxford Medical Sciences Interdivisional Research Ethics Committee (R58252/RE003).

### Defining attributes and levels

Factors expected to be important in determining preferences for prescription type (termed "attributes" in choice studies) were identified from the literature. This generated a long list of attributes that were potentially relevant both to adults and to parents considering treatment for a sick child. A survey among a small convenience sample (*n* = 22) was used to rank the attributes in order of importance (for further details, see S1 Text).

A total of 6 to 8 attributes in healthcare choice studies is generally found to be acceptable to respondents without making choices excessively complex [22–26]. All the presented situations contained all attributes. We chose to keep the attributes consistent between the adult and parent studies, for comparability, and selected 7 attributes that were (a) of common high importance to both adults and parents; and (b) of potential policy relevance. The levels chosen for each attribute were informed by a review of clinical guidelines [10,11], published literature (particularly the relevant Cochrane Review [3]), and current NHS prescribing tools and support materials [27–29] and were reviewed with 4 practising primary care physicians (who see both adults and children in the UK) and a pharmacist. Attributes, their levels, and rationale are shown in Table 1.

### Choice questions

Respondents were asked to imagine that they have an RTI, believe they might need antibiotics, and have decided to visit their primary care physician. For the parent sample, they were asked to imagine that it was their child, aged 2 years, who had the RTI. In each choice question, they were presented with a single profile describing their condition and the consultation and asked

**Table 1. Attributes and levels for the choice questions.**

| Attribute[a] | Levels | | Basis |
|---|---|---|---|
| Symptoms the person is experiencing[b] | 1: Sore throat and swollen glands 2: Chesty cough and runny nose 3: Sore throat, swollen glands, and fever 4: Chesty cough, fever, and pain on breathing | | Two upper respiratory tract symptoms, and 2 lower, to allow exploration of differences in perception of "throat" and "chest" infections. Clinical guidelines [10,11], diagnostic criteria (such as FeverPAIN [32]), and practising clinicians were consulted to identify 2 plausible levels of severity for each, identified as "minor" (1 and 2) and "serious" (3 and 4) throughout this paper. |
| How long the person has had the symptoms when they see the primary care physician | Adult: 3 days 7 days 10 days | Parent: 1 day 3 days 5 days | Durations identified from literature [3] to cover a wide yet realistic range. Durations in the parent study are shorter than in the adult one, allowing for the expected higher risk aversion of a parent seeking treatment for a sick child compared to adults seeking treatment for themselves. |
| Length of the appointment with the primary care physician | 5 minutes 10 minutes 15 minutes | | Proxy for quality of information exchange between primary care physician and patient (or parent). Levels represent plausible appointment durations; the longest appointment is intended to allow for use of tools such as TARGET patient leaflets [27] to explain treatment. |
| How much longer their usual activities will be disrupted by the illness without abx | 2 days 7 days 10 days 14 days | | Broadened from "time off work" identified in the literature review to more inclusive "usual activities." Explained as the time that people with similar symptoms usually take to feel better, which your doctor would be able to tell you. Durations based on clinical data for duration of symptoms. |
| Risk of harm from not having antibiotic treatment straight away | 1% 10% 20% | | Explained as symptoms getting worse or experiencing new symptoms. Shown as a percentage, as a graphic, and also described in words ("for every 100 patients like you, 1 would . . .") to facilitate understanding. Levels identified from review of literature on rates of complications and symptom persistence (e.g., "persistent purulent rhinitis," "not improved at follow-up," and "late recurrence") in the placebo arm of antibiotic trials in RTI [3]. |
| Risk of an adverse effect from taking abx | 1% 10% 20% | | Explained as allergy, side effects, or future resistance. Shown as a percentage, as a graphic, and also described in words ("for every 100 patients like you, 1 would . . .") to facilitate understanding. Levels identified from review of literature on adverse effects in antibiotic trials [3] and public information on rates of side effects and allergy [33]. |
| How a delayed prescription would be provided[b] | 1: Prescription + advice to delay collection of abx 2: Postdated prescription 3: Collect prescription from the practice reception at a later date | | Policy relevance: These formats have been tested in clinical trials [4] and referred to in guidelines [10], but there are no quantitative data on patient preferences. Level 1 is a standard prescription, dated on the day of the consultation, but the patient is advised to wait before collecting and taking the abx. Level 2 is a prescription with a date a number of days in the future, on, or after which the patient can collect the abx. Level 3 requires the patient to return to the practice on or after a specified date, when their prescription will be ready for them at reception. |

[a] Explanations of each attribute and its levels were provided in the survey.

[b] Categorical variable. All other attributes are treated as continuous variables.

abx, antibiotics; RTI, respiratory tract infection.

whether in that situation they would prefer an immediate or delayed prescription. The option of "no prescription" was not offered, as our aim was to understand the factors affecting preferences for a delayed prescription as an alternative to immediate prescription. We set the context of the respondent believing they might need antibiotics as a justification for choosing to visit

the doctor and to direct respondents' thinking towards their illness being potentially serious enough to be prescribed antibiotics by the doctor.

The design differs from a typical discrete choice experiment (DCE), which would usually present 2 or more alternative profiles that respondents choose between. Analysis would then use a conditional logistic regression model, reflecting that respondents have chosen a particular alternative given the finite set of profiles offered. In addition to the coefficients, the model parameters would include an alternative-specific constant; if the alternatives are labelled (e.g., one alternative is always an oral preparation and the other is intravenous), the alternative-specific constant indicates the tendency of respondents to prefer that type. In our case, each choice question presented a single profile and provided a binary choice between 2 prescription options. There is no conditionality, and the binary (1 or 0) responses are modelled using standard logistic regression. The outputs are the regression coefficients and a constant. Although less common than the multiple alternative DCE, this design has been used in other published studies in healthcare [30,31]. We chose it for this study as a better reflection of the decision-making process that our respondents would make in reality (a single illness and more than 1 option), rather than presenting several infection scenarios and requiring a choice between them.

## Survey and experimental design

The survey (provided in S2 Text) was developed by a team including 4 UK primary care physicians, a pharmacist, a patient representative, and researchers with expertise in patient and public communication regarding antibiotics. It was presented online and consisted of 3 sections:

- **Section 1** introduced RTIs, antibiotics and their use, and the concept of delayed prescription and explained the attributes and their levels.

- **Section 2** asked respondents to rank the attributes in order of importance, then provided a practice question, which was constructed using the levels most likely to indicate "immediate" prescription. The survey then presented 12 choice questions; this number of tasks has been found not to be too onerous for respondents [22–26], while maximising the amount of information generated and allowing for a good balance of representation of the levels.

- **Section 3** contained questions on respondents' experience and attitudes to antibiotics [34,35] and sociodemographic characteristics; data collected reflected our a priori expectations of parameters likely to affect respondent's choices.

The survey was tested with a convenience sample ($n$ = 10) of adults and parents identified by recommendations from the project's researchers and oversight group, to check for clarity and readability. Wording adjustments were made based on feedback from this sample.

Given the attributes and levels chosen, there are 3,888 possible profiles. Experimental design software (Ngene [36]) was used to produce the 12 choice questions in an efficient design, i.e., the algorithm generates the profiles to maximise the amount of information from respondents' choices. Constraints were applied to avoid implausible scenarios (details in S3 Text). The most efficient design was chosen: It contained no "dominant" scenarios (i.e., where, based on our a priori expectations, the levels of all attributes would increase the attractiveness of the same choice, so no trade-offs would be needed), and the levels of each attribute appeared a similar number of times. Following recommended practice [37], a preliminary design was generated and tested in a pilot among a sample of the study population (153 adults); these respondents' choices were used to refine the design for the main study, but were not included

in the final analysis. The same main study design was used for the parent study without further optimisation, allowing direct comparison.

Sample size estimation used the method of de Bekker-Grob and colleagues [38], which uses the estimated variance from the experimental design and an expected effect size. The estimate indicated that the target sample size of 800 (excluding the pilot) would be sufficient to detect coefficients of value 0.14 (equivalent to an odds ratio (OR) of 1.15) for the different levels of the symptoms attribute and 0.01 (OR equivalent 1.01) for a one-unit change in the other attributes at a 2-sided significance level of 0.05 and with power of 80%.

## Data collection

The survey was presented in English and fielded via an online panel through ResearchNow SSI (now Dynata), an online market research provider. The adult study data were collected in November 2018 and the parent data in January 2019. An online panel was chosen to enable the study to reach a large, broadly representative sample of the UK. Existing panel members were sampled to be representative of the UK population in terms of sex, age, ethnicity, and country of residence within the UK. The adult sample (inclusion criterion: age over 18 years) was recruited against quotas based on the 2011 Census for sex, age, ethnicity, and country of residence. Quotas for the parent sample (inclusion criteria: age over 18 years, parent of at least 1 child under 18 years) were based on the sociodemographic composition of parents in the "Understanding Society" dataset, a large-scale longitudinal study in the UK covering a range of social and behavioural factors [39]. Respondents were provided with on-screen information about the study and gave informed consent to participate by clicking a "Yes, I agree to take part" button. Respondent incentives were provided in the form of loyalty points, with a nominal value of approximately £1; this is a standard incentive for this provider for similar length studies.

## Analysis

Data analysis was performed in Stata (v.15SE) [40]. Choice data were analysed using a mixed-effects logistic regression model, which models the log odds of choosing delayed prescription as a linear combination of the attribute levels (Table 1). This model was chosen because it allows for heterogeneity between respondents in their inherent tendency to choose the delayed prescription option (i.e., we included a random intercept per respondent) and can incorporate respondent characteristics directly as predictors. Cluster-robust standard errors were used throughout to allow for the multiple responses by each respondent. Use of the mixed-effects logistic regression model and the respondent characteristics to evaluate were predetermined; further analyses were not prespecified.

For the categorical variables, the average predicted probability of choosing delayed prescription for each level (the marginal predicted mean) was calculated using the "margins" command in Stata [40]. This method sets the attribute to that level for all observations, keeping the other variables at their observed levels. The probability of choosing delayed prescription is then predicted for each observation using the regression model, and the mean probability is reported.

To test the robustness of the model, the main effects model was reestimated without respondents who chose delayed prescription for the practice question, chose the same option for every question, or who found the survey difficult (responded "quite difficult," "difficult," or "very difficult" to the self-reported difficulty question). To confirm whether the time and risk attributes could be appropriately represented as continuous variables with a linear relationship

with the outcome, these attributes were also modelled and plotted as categorical variables (see S4 Text).

The models were then extended to include respondent characteristics and interactions between respondent characteristics and attributes (e.g., whether males and females respond differently to specific symptoms). The attributes and characteristics to be explored in interaction models were not prespecified, but determined from the data. In initial exploratory analyses using models that allow the effect of the attributes to vary between respondents (known as random slope models), the attribute with the widest variation in its effect was symptoms. We therefore focused our analysis of interactions on this attribute.

Models were compared using a measure of how much of the variability in responses was explained by the model (McKelvey and Zavoina's Pseudo-$R^2$ [41]) and measures of goodness of fit (the Akaike and Bayesian information criteria).

Potential for reduction of antibiotic consumption in serious sore throat was estimated from the main effects model, combined with published data on levels of inappropriate prescribing, sore throat incidence, and consumption rates for delayed prescription (further information is provided in S5 Text).

## Results

### Respondent characteristics

A total of 802 adults and 801 parents completed the survey, representing completion rates of 78% and 73% of respondents who clicked on the survey link. The median time for completion was 12 minutes, and all respondents completed all questions.

Each sample was representative of the respective UK population with respect to sex, age, ethnicity, and country of residence (Table 2). Respondents reported a higher level of education than the UK population, with around 40% reporting a graduate or postgraduate qualification; this is common in other online choice studies [22,26]. The proportion of respondents reporting "none" for education was very low; such respondents were predominantly in the 65+ age group. In the adult sample, half reported that they were employed or self-employed, with a relatively high proportion of retired respondents. In contrast, 76% of the parent sample reported being employed or self-employed, reflecting the predominance of working-age people in this population. The median category for reported household income in both samples was £20,000 to £29,999, in line with the UK median income of £29,400 in 2018 to 2019 [42]. The modal income category was £10,000 to £19,999 for the adult sample and £20,000 to £29,999 for the parent sample, likely reflecting differences in the age and employment distributions.

Regarding respondents' experience and knowledge of antibiotics (Table 3), the reported number of antibiotic courses and reported awareness and experience of delayed prescription was slightly higher among parents than adults. This is probably due to their responsibility for their child(ren)'s medical needs as well as their own, hence potential for more consultations. As expected, the reported incidence of antibiotic allergy was lower in children than in adults; of note, presumed allergy may not reflect an actual allergy on testing [43–45], but the belief may nonetheless drive patients' attitudes and behaviour towards treatment. A fifth of respondents both "strongly agreed" that antibiotics are effective against bacteria and "strongly disagreed" that they are effective against viruses.

### Attribute importance

Rankings of the 7 survey attributes showed symptoms and their duration to be the most important attributes to respondents, with the format of providing the delayed prescription being the least important. The main difference between adults and parents was the ranking of

**Table 2. Respondent characteristics.**

| | | Adults (n = 802) | Parents (n = 801) | UK population[a] (%) | UK parents[b] (%) |
|---|---|---|---|---|---|
| Sex[c] | Male | 398 (50%) | 363 (45%) | 49 | 45 |
| Age[c] | 18 to 24 | 96 (12%) | 27 (3%) | 11 | 4 |
| | 25 to 34 | 128 (16%) | 238 (30%) | 17 | 30 |
| | 35 to 44 | 154 (19%) | 332 (41%) | 16 | 41 |
| | 45 to 54 | 143 (18%) | 180 (22%) | 18 | 22 |
| | 55 to 64 | 116 (14%) | 24 (3%) | 15 | 3 |
| | 65+ | 165 (21%) | 0 (0%) | 23 | |
| | Mean age (mean, SD) | 46.8 (16.93) | 38.9 (8.27) | | |
| Ethnicity[c] | White | 695 (87%) | 668 (83%) | 87 | 81 |
| | Mixed/multiple ethnic groups | 16 (2%) | 18 (2%) | 2 | 2 |
| | Black/African/Caribbean/Black British | 24 (3%) | 29 (4%) | 3 | 4 |
| | Asian/Asian British | 54 (7%) | 77 (10%) | 7 | 11 |
| | Other | 10 (1%) | 7 (1%) | 1 | 1 |
| | Prefer not to say | 3 (<1%) | 2 (<1%) | | |
| Country[c] | England | 677 (84%) | 676 (84%) | 84 | 85 |
| | Scotland | 62 (8%) | 67 (8%) | 8 | 8 |
| | Wales | 35 (4%) | 34 (4%) | 5 | 4 |
| | Northern Ireland | 28 (3%) | 24 (3%) | 3 | 3 |
| Education[d] | None | 27 (3%) | 4 (<1%) | 23 | |
| | To GCSE[e] | 184 (23%) | 184 (23%) | 14 | |
| | Post-16 | 277 (35%) | 286 (36%) | 31 | |
| | Degree or higher | 314 (39%) | 327 (41%) | 27 | |
| Employment | Employed/self-employed | 437 (54%) | 612 (76%) | 62 | |
| | Unemployed | 48 (6%) | 35 (4%) | 4 | |
| | Retired | 189 (24%) | 3 (<1%) | 14 | |
| | Long-term sick or disabled | 38 (5%) | 24 (3%) | 4 | |
| | Looking after home or family | 55 (7%) | 117 (15%) | 4 | |
| | In full-time education | 35 (4%) | 10 (1%) | 9 | |
| Household composition | With partner[f] | 502 (63%) | 651 (81%) | | |
| | Have dependent children in household | 260 (32%) | 801 (100%) | | |
| | No. of dependent children (mean, SD) | 1.8[g] (0.90) | 1.9 (0.92) | | |
| | No. of adults in household (mean, SD) | 2.1 (1.00) | 2.3 (0.93) | | |
| Gross household income | Up to £9,999 | 56 (7%) | 34 (4%) | | |
| | £10,000 to £19,999 | 165 (21%) | 123 (15%) | | |
| | £20,000 to £29,999 | 162 (20%) | 177 (22%) | | |
| | £30,000 to £39,999 | 133 (17%) | 129 (16%) | | |
| | £40,000 to £49,999 | 75 (9%) | 107 (13%) | | |
| | £50,000 to £74,999 | 111 (14%) | 118 (15%) | | |
| | £75,000 to £99,999 | 29 (4%) | 55 (7%) | | |
| | £100,000 or more | 21 (3%) | 18 (2%) | | |
| | Prefer not to say | 50 (6%) | 40 (5%) | | |

[a] Census 2011.

[b] Understanding Society 2014.

[c] Specified recruitment quotas.

[d] Highest level of qualification achieved.

[e] Includes any qualifications taken at age 16 in UK education.

[f] Includes married, civil partnership, or living with a partner.

[g] Among those with children.

GCSE, General Certificate of Secondary Education; SD, standard deviation.

**Table 3. Antibiotic experience and knowledge.**

| | | Adults (*n* = 802) | Parents (*n* = 801) | *p*-value |
|---|---|---|---|---|
| Number of antibiotic courses (past 12 months)[a] | None | 479 (60%) | 409 (51%) | 0.003 |
| | 1 | 175 (22%) | 196 (24%) | |
| | 2 | 87 (10%) | 123 (15%) | |
| | More than 2 | 61 (8%) | 73 (9%) | |
| | Mean (mean, SD) | 0.92 (2.81) | 1.03 (2.41) | |
| Recency of RTI[a] | Past month | 128 (16%) | 127 (16%) | 0.051 |
| | Past 6 months | 111 (14%) | 150 (19%) | |
| | Past year | 124 (15%) | 106 (13%) | |
| | More than a year/never | 439 (55%) | 418 (52%) | |
| Allergic to abx[a] | Yes | 120 (15%) | 58 (7%) | <0.001 |
| | No | 658 (82%) | 695 (87%) | |
| | Do not know | 24 (3%) | 48 (6%) | |
| Aware of delayed prescription[b] | Fully aware | 138 (17%) | 195 (24%) | <0.001 |
| | Partially aware | 213 (27%) | 263 (33%) | |
| | Unaware | 451 (56%) | 343 (43%) | |
| Experience of delayed prescription[c] | Once or more | 140 (17%) | 282 (35%) | <0.001 |
| Abx are effective against bacteria | Agree slightly/agree strongly | 648 (81%) | 664 (83%) | 0.625 |
| Abx are effective against viruses | Agree slightly/agree strongly | 336 (42%) | 379 (47%) | 0.096 |
| Understanding of both bacteria and viruses | Bacteria: agree strongly Viruses: disagree strongly | 149 (19%) | 166 (21%) | 0.286 |

*p*-value is for the difference between the adult and parent samples, Fisher exact test.

[a] For the parent sample, this refers to experience with any child in their care.

[b] Fully aware: aware both of the term "backup or delayed prescription" and what it is; partially aware: aware either of the term or of the approach; and unaware: not aware of either the term or the approach.

[c] Have experienced delayed prescription, i.e., been given a delayed prescription for themselves (adults) or themselves or a child (parents).

abx, antibiotics; RTI, respiratory tract infection; SD, standard deviation.

disruption, which was ranked third by adults (modal rank = 3 and mean rank = 3.91), above the risk attributes, but ranked below the risk attributes by parents (modal rank = 7 and mean rank = 4.90) (see S6 Text for details).

## Choice responses

In the choice questions, the proportion of respondents choosing a delayed prescription was very similar for adults and parents. Overall, 42% of adult choices were for delayed prescription and 44% of parents'. A total of 18% of the adult sample chose delayed prescription for the practice question and 15% of parents. Moreover, 14% of each sample chose immediate prescription for all questions, and 2% always chose delayed prescription. Over the 12 choice questions, the proportion choosing a delayed prescription ranged from 22% to 67% (see S7 Text for details).

## Choice modelling

Table 4 presents the results for the main effects analysis. The OR shows the effect of one unit of the attribute on the odds of respondents choosing the delayed prescription option. The ORs are all of the expected magnitude, giving the model face validity; i.e., OR less than 1 where we would expect an increase in the attribute to reduce the likelihood of respondents choosing the delayed prescription and greater than 1 where we would expect the likelihood to increase. An

**Table 4. Effect of attributes on preferences for delayed prescription.**

| Attribute/level | | Adults | | Parents | | p-value (heterogeneity)[a] |
|---|---|---|---|---|---|---|
| | | OR | 95% CI | OR | 95% CI | |
| Symptoms | Sore throat and swollen glands | 2.685 | 2.184 to 3.302, $p < 0.001$ | 3.707 | 2.996 to 4.588, $p < 0.001$ | 0.03 |
| | Chesty cough and runny nose | 3.735 | 2.971 to 4.695, $p < 0.001$ | 4.580 | 3.652 to 5.743, $p < 0.001$ | 0.22 |
| | Sore throat, swollen glands, and fever | 1.484 | 1.291 to 1.707, $p < 0.001$ | 1.459 | 1.271 to 1.676, $p < 0.001$ | 0.86 |
| | Chesty cough, fever, and pain on breathing[b] | 1 | - | 1 | - | - |
| Symptom duration | Per day longer | 0.935 | 0.916 to 0.956, $p < 0.001$ | 0.832 | 0.801 to 0.865, $p < 0.001$ | <0.001 |
| Appointment length | Per minute longer | 1.007 | 0.995 to 1.019, $p = 0.26$ | 0.999 | 0.986 to 1.011, $p = 0.85$ | 0.36 |
| Disruption of usual activities | Per day longer | 0.932 | 0.920 to 0.944, $p < 0.001$ | 0.961 | 0.951 to 0.971, $p < 0.001$ | <0.001 |
| Risk of harm from not starting abx | Per 1% higher | 0.983 | 0.973 to 0.993, $p = 0.001$ | 0.997 | 0.986 to 1.008, $p = 0.55$ | 0.06 |
| Risk of adverse effect from taking abx | Per 1% higher | 1.012 | 1.006 to 1.019, $p < 0.001$ | 1.017 | 1.01 to 1.024, $p < 0.001$ | 0.36 |
| Format of the delayed prescription | Advice to delay | 1.210 | 1.080 to 1.355, $p = 0.001$ | 0.933 | 0.842 to 1.035, $p = 0.19$ | 0.001 |
| | Postdated prescription | 0.945 | 0.848 to 1.054, $p = 0.31$ | 0.985 | 0.885 to 1.097, $p = 0.79$ | 0.59 |
| | Collect from practice[b] | 1 | - | 1 | - | - |
| Constant[c] | | 0.772 | 0.546 to 1.092, $p = 0.14$ | 0.613 | 0.433 to 0.867, $p = 0.006$ | - |

[a] p-value for heterogeneity is the p-value for the interaction term between each attribute and the sample (adult or parent) from a combined model including both samples ($n = 1603$). A low p-value (e.g., <0.05) indicates the attribute has a different effect on choices in adult and parent samples.

[b] Reference level for the categorical variables. The OR for each level shows the ratio of the odds of choosing delayed prescription, relative to the reference level (for a unit increase in continuous variables).

[c] Constant reflects the probability (on the logit scale) of choosing delayed over immediate prescription at the reference value of all other factors.

abx, antibiotics; CI, confidence interval; OR, odds ratio.

OR of 1, or where the 95% confidence interval (CI) includes the value of 1, indicates no evidence of an effect of the attribute on respondents' choices.

Compared to the more serious lower respiratory tract symptoms (chesty cough, fever, and pain on breathing), respondents were more likely to choose the delayed prescription for all of the other symptoms described (high ORs), particularly the minor symptoms. They were most likely to choose the delayed prescription for the cold-like symptoms of chesty cough and runny nose. Respondents were more likely to choose delayed prescription (OR > 1) if there was an increased risk of adverse effects from treatment. Respondents were less likely to choose delayed prescription (OR < 1) if the symptoms had been present for longer or if they were expected to experience disruption for longer. Adults were less likely to choose delayed prescription if there was an increased risk of harm from delaying treatment, but there was no evidence of an effect of this attribute for parents. There was no evidence that length of the appointment with the primary care physician affected preferences. The format of the delayed prescription had an effect on adults' choices: Compared to having to return to the practice to collect the prescription from reception, respondents were more likely to choose delayed

prescription if they were handed a prescription and advised to delay starting treatment, but not if they would be given a postdated prescription. In contrast, there was no evidence that the format affected parents' preferences.

The adult model suggests that the impact of a day's duration of illness before the appointment had a similar effect on prescription preferences to a day's subsequent disruption of usual activities (ORs very similar). In contrast, among parents, the duration of their child's illness had a more marked effect on their choices, with a smaller effect of the expected disruption to their joint usual activities. For parents, an additional day that their child had experienced symptoms had over 4 times the effect on their choice as an additional day of future disruption (17% reduction in the odds of choosing delayed prescription, per additional day, compared to 4%).

Predicting from the model, the probability of choosing delayed prescription in the adult sample was 0.30 (95% CI 0.28 to 0.33, $p < 0.001$) for chesty cough, fever, and pain on breathing, rising to 0.53 (0.50 to 0.56, $p < 0.001$) for the minor cough symptoms (chesty cough and runny nose) and to 0.37 (0.34 to 0.39, $p < 0.001$) and 0.47 (0.44 to 0.50, $p < 0.001$) for the serious and minor sore throat symptoms, respectively. Probabilities for parents choosing delayed prescription for a sick child were slightly higher than for adults for the minor sore throat symptoms.

The adult main effects model explained 12% of the variation in respondents' choices based only on the attributes and 37% when respondent-level variability was included (12% and 38%, respectively for the parent sample), i.e., around two-thirds of the explained variation was due to differences in individuals' tendency to choose delayed prescription, rather than the attributes. The models were robust to the exclusion of respondents who always chose the immediate or the delayed prescription, chose delayed prescription in the practice question, or found the survey difficult.

## Respondent characteristics and interactions

A second pair of models incorporated respondent characteristics and their interactions with the symptoms attribute where those interactions improved model fit (see S8 Text for details). These models had slightly improved fit compared to the main effects model, and the proportion of variation explained based on the attributes (excluding individual-level variability) increased to approximately 20%.

The effects of the interaction terms on the probability of choosing delayed prescription are illustrated in Fig 1. Taking sex as an example, the effect of the more serious symptoms including fever on preferences was similar for males and females. However, male and female preferences differed significantly on the 2 minor symptoms; the probability of females choosing delayed prescription was 0.51 (0.47 to 0.55, $p < 0.001$) for sore throat and swollen glands and 0.62 (0.58 to 0.66, $p < 0.001$) for chesty cough with runny nose compared to 0.43 (0.40 to 0.47, $p < 0.001$) and 0.45 (0.41 to 0.48, $p < 0.001$) for males for these conditions. Similar patterns of higher probability of choosing delayed prescription for the minor symptoms (particularly cough) were observed for respondents who were knowledgeable about antibiotics and those who had not been prescribed antibiotics in the past year. These observations hold for both adult and parent samples.

In the adult sample, responses to the symptoms differed by age. The youngest age group in the adult sample showed little difference in probability of choosing delayed prescription across all the different symptoms. With increasing age, respondents showed an increasing tendency to choose delayed prescription for the minor symptoms compared to younger respondents. In the parent sample, we did not find evidence of an interaction between symptoms and

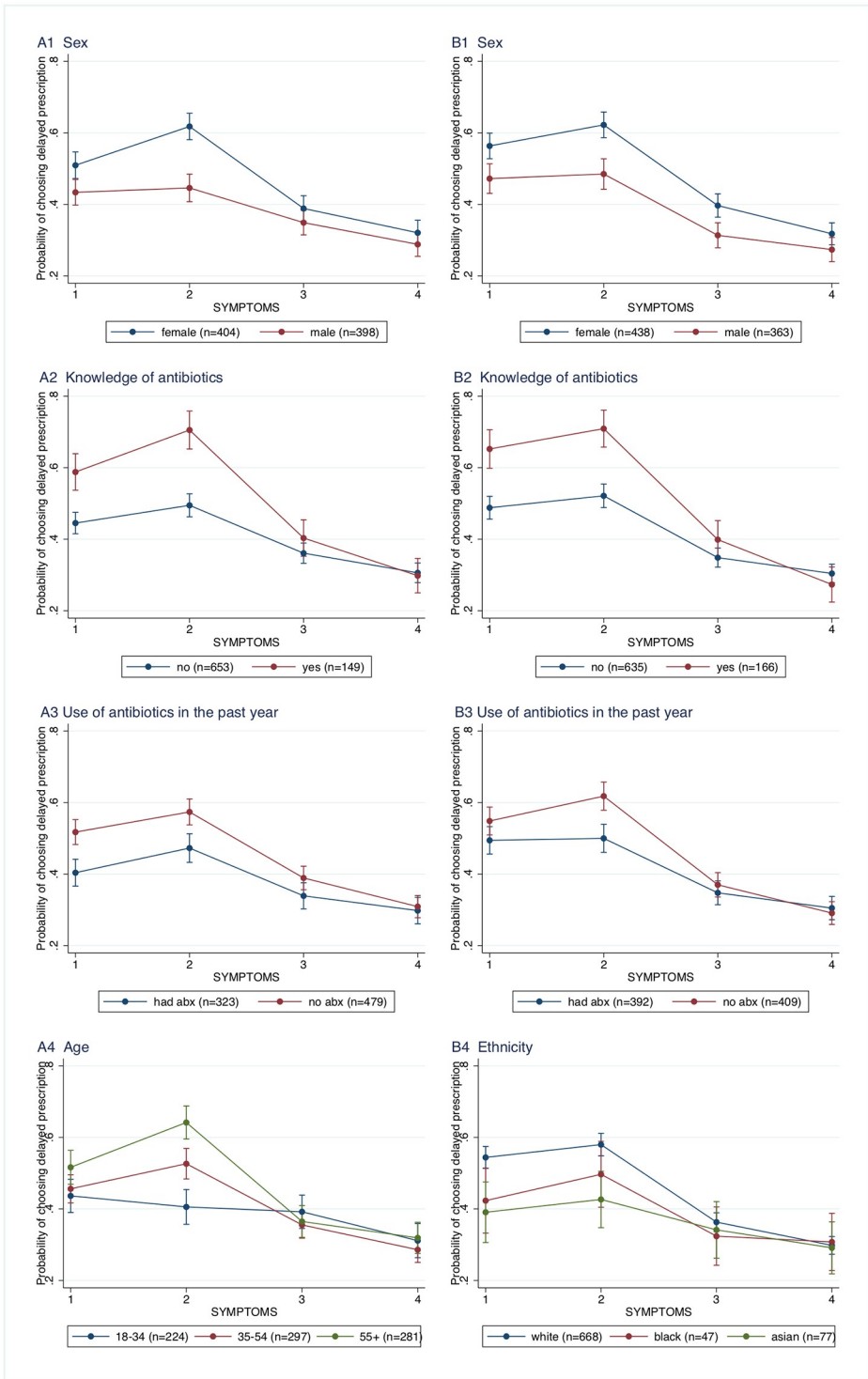

**Fig 1. The probability of choosing delayed prescription varies by respondent characteristics across symptom types. (A)** Adults. **(B)** Parents. Symptoms: 1 –sore throat and swollen glands; 2 –chesty cough and runny nose; 3 –sore throat, swollen glands, and fever; and 4 –chesty cough, fever, and pain on breathing. Bars indicate 95% CIs. Ethnicity: White includes "white," "other," and "prefer not to say," and black includes "black" and "mixed ethnicity." abx, antibiotics; CI, confidence interval.

respondent (i.e., parent) age; however, there was a significant interaction between symptoms and ethnicity, with Asian parents being less likely than white parents to choose the delayed prescription for a child with minor symptoms.

## Discussion

Our findings suggest that the most important factors for the UK general public in acceptance of delayed prescription in RTIs are the severity and type of symptoms experienced, and, particularly for parents considering a sick child, the duration of those symptoms. Parents gave lower weighting to the risk of delaying antibiotics than to the risk from immediate antibiotic treatment and to future disruption of their usual activities due to their child's illness compared with adults. Females, older adults, those with a good understanding of antibiotics, and those who had not been prescribed antibiotics recently were most amenable to delayed prescription. These groups were particularly likely to choose delayed prescription for the less serious conditions described, particularly for cold-like symptoms (chesty cough and runny nose).

The acceptance of delayed prescription was highest for cold-like symptoms of a cough and runny nose, suggesting that educational messages that antibiotics are not needed for colds have had some effect (at least in older individuals); this is consistent with a recent European survey showing that 79% of UK adults know that antibiotics are not effective in treating colds [46]. The lower acceptance for sore throat suggests potential for educational interventions to clarify the position for sore throat treatment; i.e., that antibiotics are rarely indicated, just as for coughs and colds. However, even assuming the current level of acceptance found in this study (37% probability of choosing delayed prescription for sore throat with swollen glands and fever, among adults overall), the public appear to be more open to delayed prescription in serious sore throat than current prescribing data suggest. Based on our results, combined with estimations of the level of inappropriate prescribing in sore throat [19] and the prevalence of delayed prescription [6], we estimate that an additional 9% to 12% of current sore throat prescriptions could be replaced by delayed prescriptions. Assuming that 69% of these delayed prescriptions resulted in the prescription never being taken [3], this would result in, conservatively, over half a million fewer antibiotic prescriptions being initiated per year across the UK for sore throats (for estimation details, see S5 Text).

The preferred format for delayed prescription was to provide a prescription with advice to delay collecting the antibiotics. A trial of the various formats found no evidence for differences in antibiotic use or symptom control [4]. Hence, from the patient's perspective, our study finds no support for recommending the formats where the patient's choice to initiate antibiotics is more tightly defined.

To our knowledge, this is the first study in the UK to attempt to quantify the trade-offs that members of the public make between the factors determining preferences for delayed prescription. A 2014 survey on public attitudes to delayed prescription found similar levels of awareness. However, in contrast to our findings, they reported that women and older age groups were more likely to be opposed to delayed prescription [35]. Our approach of presenting specific scenarios may have provoked responses that more closely reflect actual decision processes compared to survey questions asking whether respondents are in principle in favour of delayed prescription. Other surveys of public beliefs about antibiotics emphasise the importance of symptoms and their duration in the decision to seek or take antibiotics and identify similar lack of understanding of the role of antibiotics among some population subgroups [34,47]. For example, a large-scale European survey found that females and those aged over 25 years were more likely to be knowledgeable about antibiotics [46], which is consistent with our findings. This suggests that our findings may be generalisable to primary care prescribing for RTIs in

other jurisdictions. However, for other conditions (such as urinary tract infections), acceptance of delayed prescription may be affected by factors not tested in this study.

The parent model suggests that parents do not have a high level of concern with risk of harm due to delaying antibiotic treatment for their child. This finding is intriguing, as we might expect parents to be anxious to get active treatment for their child's illness, and it is not consistent with the parents' ranking of the attributes. This response may reflect the fact that parents tend to consult earlier with a child, hence with less serious symptoms; the risk from delaying treatment may appear less relevant at this early stage. Parents were also more likely to choose immediate antibiotics for prolonged symptoms, and it may be that duration of symptoms is acting as a proxy for the risk from delaying treatment.

The adult and parent samples differed in the relative importance given to future disruption of usual activities, with parents putting greater weight on how long their child had been ill. This is consistent with the parents' lower ranking of the disruption attribute and may be explained by the difference in the actors involved; particularly, in this hypothetical context, parents may feel that their own convenience should not determine their child's exposure to antibiotics. This observation may be sensitive to employment status, particularly if respondents have some flexibility in their work schedule; part-time work had a small effect when included alongside just the main effects, but no significant effect in the full model, probably due to the strong correlation between part-time work and sex.

Similarly, parents' choices did not seem to be affected by the format of the delayed prescription. The larger effects of the symptoms and duration attributes among parents suggest that they focused on these attributes and perhaps paid little attention to the format. This may not reflect their reaction in reality when asked to return to the surgery to collect the prescription.

In addition to age and sex, we identified 2 groups of respondents who were more likely to choose delayed prescription: those defined as knowledgeable about antibiotics and those who reported they had not been prescribed antibiotics in the past year. The "knowledge of antibiotics" group understood that antibiotics are effective against bacteria but not viruses. We suggest that their knowledge is not limited to this fact, but reflects a broader understanding of antibiotics and infections, and, perhaps, higher health literacy or interest in health in general. This group tended to have a higher level of education and were less likely to choose "do not know" in the health and antibiotics questions. The group who had not been prescribed antibiotics in the past year may be generally "well," reflected in the lower reported incidence of recent RTIs in this group and their low awareness of delayed prescription. Alternatively, they may have a lower tendency to consult a primary care physician.

Our identification of population subgroups who are more amenable to delayed prescription and their responses to the specific symptom types may present an opportunity for primary care physicians who wish to increase appropriate use of delayed prescription, but who are concerned about patient acceptance. Our findings indicate that such patients would need to be reassured that their symptoms are appropriate for delayed prescription, particularly if the symptoms are similar to the "more serious" versions in this study with the presence of fever. In particular, patients may need careful explanation for sore throat symptoms, where the acceptability of delayed prescription is lower. In addition, parents may need particular reassurance about the typical duration of a sore throat, if they are concerned about how long their child has been ill.

Our findings also suggest that intervention is needed to increase acceptance of delayed prescription more broadly among the public. While this could be achieved by primary care physicians during consultations, by eliciting and addressing patient concerns, this approach may not be an effective use of limited time if patients are starting from a low knowledge base. An alternative approach may be to increase targeting of broader educational interventions to

population subgroups where acceptance of delayed prescription is lower, such as younger age groups and males. The aim would be to increase understanding of the role of antibiotics, such that these patients are then more amenable to delayed prescription when they subsequently present with a relevant infection in primary care.

The study is limited by its hypothetical nature, so responses may not reflect the way patients or parents would behave in reality. The concept of delayed prescription in itself may be difficult to understand, and patients may respond differently in a consultation, when they are unwell and worried. In addition, in some of these scenarios, respondents might not have consulted a doctor at all, and, for the conditions where they would consult, the acceptability of delayed prescription may be lower than seen here. Further, to avoid excessive respondent burden, the number of attributes described was constrained, so attributes of potential relevance to adults and/or parents may have been omitted. Despite this, the survey may have been complicated for respondents to understand or not reflected their experience, and they may have used simplifying heuristics to make the choices easier, such as focusing on specific attributes and ignoring others, which introduces unexplained heterogeneity into the results. For the parent sample, we also asked them to think back to when they had a 2-year-old child, which may have been challenging for those with older children. However, age of children had little effect on the parent models, so closeness to the scenario appears not to have affected choices systematically. A revealed preference study, where data are collected on respondents' actual preferences when offered a delayed prescription, would complement our findings.

The study sample reflected the UK population in being predominantly white and living in England; the respondents were also more educated on average than the population, and the majority of parents lived with a partner. This is a common observation in online surveys. However, our findings may therefore not reflect reality in neighbourhoods with, for example, a higher proportion of ethnic minorities, migrants, single-parent families, or particular challenging socioeconomic situations. Further research is needed to understand these groups' views and response to future interventions.

The study design did not allow for evaluation of interactions between the attributes, such as between the symptoms and the expected disruption (a longer duration of disruption may be less acceptable if the symptoms are more severe). The design was optimised for the estimation of the main effects of each attribute, but not for 2-way or more complex interaction terms. We did attempt some exploratory analyses, but were unable to find strong evidence for interactions, at least in part because these analyses were complicated by collinearity between the interaction terms. Further, in common with many choice experiments using multiattribute profiles, our 12 choice questions are a small sample of all possible profiles (although some of those would be implausible, so can be ruled out). Experimental design software was used to optimise the choice of profiles selected for the study. However, specific interactions and a broader sample of profiles could both be investigated in future studies by using an incomplete block design; a larger number of choice questions are generated, but each respondent is only presented with a subset of them [37].

The choice question in our study was between an immediate or delayed prescription. We did not include a "no prescription" option, as our study question related specifically to delayed prescription as a mechanism to reduce unnecessary antibiotic consumption compared to immediate prescription. We framed the choice question to be internally consistent with the choices offered, by telling respondents "you think you might need antibiotics." Indeed, the possibility of taking no antibiotics is implicit in the delayed prescription approach. However, in reality, not prescribing would be an option for the prescriber. Including a "no prescription" alternative may have led to different choices in the study. Intuitively, these might be expected to be a subset of the "delayed prescribing" choices. It is possible that despite our framing of the

question, respondents did not want antibiotics at all in some cases, so our study would have underestimated respondents' openness to not being given an antibiotic prescription. It may also be that the presence of a "no prescribing" option could have made the delayed prescription more appealing by making it appear as an intermediate option on an extended scale of prescribing options. However, in both of these cases, our results represent a more conservative estimate of the acceptance of alternatives to immediate prescribing.

Our study focuses on the preferences of the public, and the choice scenarios do not consider the behaviour of the prescriber. Further work is needed to understand the drivers for use of delayed prescription among primary care physicians.

In conclusion, we found that symptoms and their duration are the strongest drivers of acceptance of delayed over immediate antibiotic prescription and identified subgroups of the population who are particularly amenable to delayed prescription for minor infections, particularly cold-like symptoms. Parents considering a sick child had a similar overall likelihood to the adult sample of choosing delayed prescription, but placed higher weight on duration of symptoms and less on future disruption of their activities due to caring for their child. Our findings could help to reduce consumption of antibiotics in primary care by encouraging primary care physicians to increase their use of delayed prescription in those groups who are more open to this approach and to specifically address concerns such as illness duration. Educational interventions to improve understanding of antibiotics could target those who are less amenable to delayed prescription and focus on the (lack of) role of antibiotics in sore throat.

## Supporting information

**S1 Text. Identification and selection of attributes.**
(PDF)

**S2 Text. Survey instrument (adult version).**
(PDF)

**S3 Text. Constraints placed on the experimental design.**
(PDF)

**S4 Text. Modelling continuous variables as categorical.**
(PDF)

**S5 Text. Estimation of the impact on antibiotic prescribing.**
(PDF)

**S6 Text. Attribute importance.**
(PDF)

**S7 Text. Respondent choices.**
(PDF)

**S8 Text. Models incorporating respondent characteristics and interactions.**
(PDF)

**S1 Checklist. ISPOR Checklist for conjoint analysis applications in health.**
(PDF)

## Acknowledgments

We thank all respondents who took part in our survey. The STEP-UP team consists of the following: Philip E Anyanwu, Aleksandra Borek, Nicole Bright, James Buchanan, Christopher

Butler, Anne Campbell, Ceire Costelloe, Benedict Hayhoe, Alison Holmes, Susan Hopkins, Azeem Majeed, Monsey McLeod, Michael Moore, Liz Morrell, Koen B Pouwels, Julie V Robotham, Laurence S J Roope, Sarah Tonkin-Crine, Ann Sarah Walker, Sarah Wordsworth, Carla Wright, and Anna Zalevski.

**Disclaimers**: The views expressed are those of the authors and not necessarily those of the NHS, the NIHR, the Department of Health, or PHE.

## Author Contributions

**Conceptualization:** Liz Morrell, James Buchanan, Laurence S. J. Roope, Koen B. Pouwels, Christopher C. Butler, Benedict Hayhoe, Sarah Tonkin-Crine, Monsey McLeod, Julie V. Robotham, Alison Holmes, A. Sarah Walker, Sarah Wordsworth.

**Data curation:** Liz Morrell.

**Formal analysis:** Liz Morrell.

**Funding acquisition:** Julie V. Robotham, Alison Holmes, A. Sarah Walker, Sarah Wordsworth.

**Methodology:** Liz Morrell, James Buchanan, Laurence S. J. Roope, Koen B. Pouwels, Julie V. Robotham, A. Sarah Walker, Sarah Wordsworth.

**Supervision:** A. Sarah Walker, Sarah Wordsworth.

**Writing – original draft:** Liz Morrell.

**Writing – review & editing:** Liz Morrell, James Buchanan, Laurence S. J. Roope, Koen B. Pouwels, Christopher C. Butler, Benedict Hayhoe, Sarah Tonkin-Crine, Monsey McLeod, Julie V. Robotham, Alison Holmes, A. Sarah Walker, Sarah Wordsworth.

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
