## [Decision Letter · Decision Letter 0]

31 Jan 2020

Dear Dr. Morrell,

Thank you very much for submitting your manuscript "Delayed and immediate antibiotic prescriptions in primary care: what does the UK public prefer?" (PMEDICINE-D-19-04469) for consideration at PLOS Medicine. 

Your paper was evaluated by an academic editor with relevant expertise and sent to independent reviewers, including a statistical reviewer. The reviews are appended at the bottom of this email and any accompanying reviewer attachments can be seen via the link below:

[LINK]

In light of these reviews, we will not be able to accept the manuscript for publication in the journal in its current form, but we would like to invite you to submit a revised version that fully addresses the reviewers' and editors' comments. You will appreciate that we cannot make a decision about publication until we have seen the revised manuscript and your response, and we expect to seek re-review by one or more of the reviewers. 

We hope to receive your revised manuscript by Feb 21 2020 11:59PM. Please email us (plosmedicine@plos.org) if you have any questions or concerns.

Please let me know if you have any questions. Otherwise, we look forward to receiving your revised manuscript in due course. 

Sincerely,

Richard Turner, PhD

rturner@plos.org

Please supply a non-author contact for inquiries about access to study data, as required by PLOS' data policy.

Please adapt the title to better match journal style. We suggest "Public preferences for delayed or immediate antibiotic prescriptions in UK primary care: a discrete choice experiment". 

Please quote the dates over which the study was done in the abstract.

In the abstract and elsewhere in the paper, please add p values alongside 95% CI, where available. 

In the final sentence of the "methods and findings" subsection of your abstract, summarizing the study's main limitations, we ask you to add 1-2 further limitations (e.g., the issue of unclear relevance of your findings to other populations). 

After the abstract, we will need to ask you to add a new and accessible "author summary" section in non-identical style. You may find it helpful to consult one or two recent research papers published in PLOS Medicine to get a sense of the preferred style. 

Early in the methods section, please state whether the study had a protocol or prespecified analysis plan, and if so attach the document(s) as a supplementary file (referred to in the methods section). Please highlight any analyses that were not prespecified. 

At line 332, please soften the wording slightly to: "Our findings suggest that the most important factors ...". 

Throughout the paper, please quote p values as "p<0.001" or exact values. 

Throughout the paper, please provide greater emphasis on reported observations, e.g., at line 203 "... of the parent sample reported that they were employed or self-employed ...". 

Please substitute "sex" for "gender" as appropriate throughout the text. 

Throughout the article, please adapt reference call-outs to the following style: "... Europe [6,7,12].".

In your reference list, please ensure that journal names are abbreviated consistently, e.g., "Cochrane Database Syst Rev." for reference 4.

Please add a completed checklist for the most appropriate reporting guideline, which we suspect will be STROBE, referred to in the methods section. In the checklist, please refer to individual items by section (e.g., "Methods") and paragraph number rather than by page or line numbers, as the latter generally change in the event of publication. 

Comments from the reviewers:

*** Reviewer #1: 

[See attachment]

Michael Dewey

*** Reviewer #2: 

This is a generally well written DCE on an important topic. The below comments are meant to be constructive.

I have to say I struggled with the meaning and logic of some of the design attributes. For example, the attribute about length of interruption is strange, as this will not be known with certainty? Therefore how can it be specified in advance. Also wouldnt this be correlated with symptoms? Risk of harm is very ambiguous and a resistance as terms / concepts - what could they imply in terms of impact on health? No or little information is given about this. Also for a DCE to work, participants must trade off attributes. So there must be an upside and downside to a choice. Here I struggled with the design in so much that I am not clear what is meant by the 'risk of harm' or adverse effect of taking an AB, or how often they occur. 20% seems very high - is this plausible?

This is possibly a reflection of my understanding but I dont entirely understand the design. Normally in a DCE different profiles are assembled for different options eg A v B, but here it isnt the case. The A v B (or labels) are still there but the attributes are the same across the profiles. For example, someone is going to be ill for 14 days irrespective of choice (In the example)? How then is the independent importance of this attribute determined, given it hasn't been rotated? Additionally, in this sense length of symptom for example is not a design attribute (of the AntiB itself), its background factor. I would suggest that as a minimum the design / analysis approach should be better explained.

Given this is effectively a labelled design, is there a constant in the model to reflect the independent preference for one of the two option (ie the alternative specific constant)?

It strikes me as important that no prescription was omitted as an option. If it had been included, could it have impacted the results. That is, if the alaternative was not receiving anything, then pressuable a delayed prescrption would be more appealing? 

I would suggest an OR of 0.997 is equivalent to 1 for all practical purposes, hence I wouldn't describe the attribute as having a modest impact.

I am vey surprised that the format of the delayed prescription didn't have any impact. I would have thought going back to the practice would have been seen as annoying and unnecessary (on the premise that people usually highly value their time)?

The paper assumes people who receive a delayed prescription do not take them if not required. But surely an issue with this approach is that they might be kept and used a later date (without first seeking medical advice)?

Even a person was willing to delay, it might not be medically appropriate, therefore where does this leave the results?

How was the sample size estimate calcuated?

Are respondents paid to be complete the survey, if so what biases could this introduce?

*** Reviewer #3: 

Comments to the Editor

Overall this is an interesting study on a relevant topic, i.e. the acceptability of delayed prescription and the underlying drivers for acceptance by the general public.

This study complements nicely earlier work done on the potential role of delayed prescriptions in the decrease of outpatient antibiotic use for common pathologies. Its results may serve also as valuable background information information to feed into educational programmes and/or more targeted use of delayed prescriptions.

In general the study appears to have been designed, planned and carried out meticulously. However, being a clinician, I am not fully confident with all the statistical methods used. Please make sure that someone with more epidemiologicl/methodological experience revises the manuscript as well for soundness of methodology and analysis used.

Comments to the authors

Overall this is an interesting study on a relevant topic, i.e. the acceptability of delayed prescription and the underlying drivers for acceptance by the general public.

This study complements nicely earlier work done on the potential role of delayed prescriptions in the decrease of outpatient antibiotic use for common pathologies. Its results may serve also as valuable background information information to feed into educational programmes and/or more targeted use of delayed prescriptions.

In general the study appears to have been designed, planned and carried out meticulously.

Methods

This section is very detailed and convincing that the authors designed their study very meticulously. A few remarks:

-survey instrument: among the survey developers, I did not notice a pharmacist nor a pediatrician. I think his is a pity as both may play an important role in the creation or carrying out of delayed prescriptions

-population surveyed: in both groups there is a clear predominance (87/83%) of white people from British origin. Also, 81% of the respondents lived with their partner, and on average the population was more educated than average. This may not reflect the reality in many poor inner city neighbourhoods, with higher percentages of ethnics groups in challenging socio-economical situations, single parents etc… Migrants may have yet very different views and understanding on the use of antibiotics. This discrepancy may cause a considerable bias when it comes to drawing conclusions and designing a strategy for further interventions. Perhaps this study should be repeated (maybe with other methodologies) targeting these risk groups.

Results

Table 2b: 15% of the adults mention a problem of 'allergy'. I am surprised the authros did not mention this in their discussion, as it is well known that presumed allergy is very often not real, but is an important reason for inappropriate use of second line antibiotics with even more side effects and selection of resistance (e.g. fluoroquinolones).

Ln 272: the difference between 'delayed' and a 'post-dated prescription' is not fully clear to me as a reader. Please define clearly and add to the method's section

Ln 281: the findings from Figure 1 appear to be discordant with the Odd's ratio's mentioned in Table 3. Please explain how you obtained these results and how to explain these differences.

Discussion

Ln 355 Please explain how you obtained the estimate that an additional 9-12% of prescriptions can be turned into delayed prescriptions?

Ln 416 regarding parents starting from a low knowledge basis: see my earlier comment on the risk for biased population in this study, towards more educated persons/parents from higher socio-economical classes. In view of the current economical and societal changes, it will be important to retrieve data from other, more vulnerable/hard to reach groups in our societies as well

***

[LINK]

---

## [Decision Letter · Decision Letter 1]

24 Jan 2021

Dear Dr. Morrell,

Thank you very much for re-submitting your manuscript "Public preferences for delayed or immediate antibiotic prescriptions in UK primary care: a choice experiment" (PMEDICINE-D-19-04469R1) for consideration at PLOS Medicine. We do apologize for the long delay in sending you a decision. 

I have discussed the paper with editorial colleagues, and it was also seen by one previous reviewer and two new reviewers. I am pleased to tell you that, provided the remaining editorial and production issues are fully dealt with, we expect to be able to accept the paper for publication in the journal.

[LINK]

Please let me know if you have any questions, and we look forward to receiving the revised manuscript shortly.   

Sincerely,

Richard Turner, PhD

rturner@plos.org

Requests from Editors:

Please make that "has been shown ..." in the second line of the abstract. 

Please quote summary details on respondent demographics in your abstract, and the survey response rates. 

In the abstract and throughout the text, please quote p values alongside 95% CI, where available. 

In your abstract, please quote quantitative data, including 95% CIs and p values, for the following results: “Respondents were less likely to choose delayed prescription with increasing duration of illness. Probabilities were similar for parents considering treatment for a child. However, parents differed from the adult sample in giving higher weighting to the duration of illness, and lower weighting to future disruption of usual activities. Females were more likely to choose a delayed prescription than males, and were particularly likely to do so for minor cough symptoms.”

Please condense the discussion of study limitations in your abstract to a single sentence, concluding the "Methods and findings" subsection, e.g., "Study limitations include the hypothetical nature of the choices offered, which may not reflect real-life behaviour; the absence of an option of no prescription; and the possibility that study respondents may not be representative of the population as a whole or of groups within it.".

Please adapt the first sentence of the "Conclusion" subsection of our abstract to begin "In this study, we found that delayed prescription ..." or similar. 

Please make that "... (lack of) a role ..." in the author summary. 

Please condense the discussion of the study aim and design at the end of the Introduction section. 

As discussed in your cover letter, please state explicitly early in your methods section that the study did not have a prespecified analysis plan.

In your methods section, please add a brief discussion on the question of informed consent for study respondents. 

In the discussion section, where you mention that you "quantify the trade-offs", please amend this to "attempt to quantify" or "estimate the trade-offs", or similar. 

Please remove information on funding from the Acknowledgements section at the end of your main text. This information will appear in the article metadata, via entries in the submission form. 

Please ensure that all references meet journal format. For example, "BMJ" will suffice as the journal name for reference 4, and this citation may be lacking some access information. 

Please check that the URL provided for reference 45 functions, and add accessed dates to the web references. 

Please ensure that journal names are abbreviated consistently. 

Comments from Reviewers:

*** Reviewer #1: 

The authors have addressed all my points

Michael Dewey

*** Reviewer #4: 

Thank you for the opportunity to contribute to the review for this article, which I found to be well-written and describing an interesting and important topic. Thoughtful and thorough comments have already been provided by three reviewers, and in my opinion the authors have addressed the points raised and the article is suitable for publication in PLoS Medicine.

I have the following minor points:

- Agreed that a blocked design would have been interesting to better explore the space and potentially better opportunities for interactions - this seems worth mentioning as a limitation but not a "fatal flaw"

- Multiple reviewers pointed out the absence of a 'no prescription' option and the authors address this in the discussion. However, I would argue that in this context, there may not be need for such an option - presumably implicit in the delayed prescription is the potential for the patient (or child)'s condition to improve spontaneously and allow them to spare the antibiotics. In that case, it would seem that a delayed prescription already carries the potential for 'no medication' which would be functionally equivalent to 'no prescription', but leaves this decision and its timing to the patient and in that case could be seen as a more empowering/patient-centric approach than an immediate prescription

- Reviewers noted the scale of the variables for reporting odds ratios - of note, one option might be to report values for, e.g. 5% or 5 minutes etc., to more naturally track with the differences between attribute levels. Alternatively (or in addition to), I have contributed to DCE studies in the past where we have used barplots to communicate the probability of making a particular selection across several representative collections of attributes and have had good feedback in that as clarifying interpretation of the regression results.

*** Reviewer #5: 

Many thanks for the opportunity to review this revised manuscript. The study represents a valuable addition to the literature and advances our knowledge of preferences for delayed versus immediate antibiotic prescriptions. The authors have responded to the peer-review comments and have amended the manuscript appropriately. The theoretic reasoning for immediate versus delayed, and the absence of a "no prescription" option - is adequately described, with appropriate discussion around the limitations of this design. The experiment is now clearly described. The use of the term 'delayed' and use of 'back-up' in parentheses, or lay versions - was justified and meets with the understanding of a wide ranging readership. The added clarity in the manuscript allows readers to interpret the findings based on the clearly described hypothetical scenario, limitations are acknowledged and interpretation is supported by the transparency in reporting. The manuscript appears to have undergone appropriate revision to address the reviewers comments, and improving the clarity of a valuable study.

***

[LINK]

---

## [Editor Report · Decision Letter 2]

15 Jul 2021

Dear Dr Morrell, 

On behalf of my colleagues and the Academic Editor, Dr Low, I am pleased to inform you that we have agreed to publish your manuscript "Public preferences for delayed or immediate antibiotic prescriptions in UK primary care: a choice experiment" (PMEDICINE-D-19-04469R2) in PLOS Medicine. We do apologize for the long delay in sending you a response. 

Prior to final acceptance, please revisit the section of your abstract where you present different preferences expressed by parents and adults: "... higher weighting (OR 0.83 ...) ... lower weighting (OR 0.96 ...)" is potentially confusing, and we suggest amending this to: "... higher weighting to the duration of illness (OR 0.83 [95% CI 0.80-0.87] vs 0.94 [95% CI 0.92-0.96] p for heterogeneity <0.001) reflecting a lesser preference expressed by parents for a delayed prescription ..." or similar.

PRESS

Sincerely, 

Richard Turner, PhD 

rturner@plos.org